# Efficiency of antioxidant Avenanthramide-C on high-dose methotrexate-induced ototoxicity in mice

**Alphonse Umugire**[1,2], **Youngmi Choi**[1], **Sungsu Lee**[1], **Hyong-Ho Cho**[1,2]*

**1** Department of Otolaryngology-Head and Neck Surgery, Chonnam National University Medical School and Chonnam National University Hospital, Gwangju, Korea, **2** Department of Biomedical Science, College of Medicine, Chonnam National University Graduate School, BK21 PLUS Center for Creative Biomedical Scientists at Chonnam National University, Gwangju, Korea

* victocho@hanmail.net

**Data Availability Statement:** All relevant data are within the paper and its Supporting information files.

**Funding:** This research was supported by a grant (NRF-2020R1A2C1007473) from the Basic

## Abstract

Methotrexate (MTX) has been used in treating various types of cancers but can also cause damage to normal organs and cell types. Folinic acid (FA) is a well-known MTX antidote that protects against toxicity caused by the drug and has been used for decades. Since hearing loss caused by MTX treatment is not well studied, herein we aimed to investigate the efficiency of the antioxidant Avenanthramide-C (AVN-C) on high-dose MTX (HDMTX) toxicity in the ear and provide insights into the possible mechanism involved in MTX-induced hearing loss in normal adult C57Bl/6 mice and HEI-OC1 cells. Our results show that the levels of MTX increased in the serum and perilymph 30 minutes after systemic administration. MTX increased hearing thresholds in mice, whereas AVN-C and FA preserved hearing within the normal range. MTX also caused a decrease in wave I amplitude, while AVN-C and FA maintained it at higher levels. MTX considerably damaged the cochlear synapses and neuronal integrity, and both AVN-C and FA rescued the synapses. MTX reduced the cell viability and increased the reactive oxygen species (ROS) level in HEI-OC1 cells, but AVN-C and FA reversed these changes. Apoptosis- and ROS-related genes were significantly upregulated in MTX-treated HEI-OC1 cells; however, they were downregulated by AVN-C and FA treatment. We show that MTX can cause severe hearing loss; it can cross the blood–labyrinth barrier and cause damage to the cochlear neurons and outer hair cells (OHCs). The antioxidant AVN-C exerts a strong protective effect against MTX-induced ototoxicity and preserved the inner ear structures (synapses, neurons, and OHCs) from MTX-induced damage. The mechanism of AVN-C against MTX suggests that ROS is involved in HDMTX-induced ototoxicity.

## Introduction

Methotrexate (MTX), once recognized as amethopterin, is a chemotherapeutic agent that also acts as an immunosuppressant [1]. MTX is an antifolate and antimetabolite with

Science Research Program through the National
Research Foundation of Korea, funded by the
Ministry of Education, Science and Technology. It
was also supported by a grant from the Chonnam
National University Hospital Biomedical Research
Institute (BCRI20038). The funders had no role in
study design, data collection and analysis, decision
to publish, or preparation of the manuscript.

**Competing interests:** The authors have declared
that no competing interests exist.

immunomodulatory activity against a variety of inflammatory conditions. It inhibits folate
metabolism through the suppression of dihydrofolic acid reductase, thereby preventing purine
and pyrimidine synthesis and reducing DNA and RNA synthesis [2]. Its pharmacokinetics
and potential toxic effects, such as nephrotoxicity and hepatotoxicity, are also well understood
[3]. MTX has been effectively and extensively used for treating various types of cancers [4].
MTX induced the production of reactive oxygen species (ROS) in monocytes and cytotoxic T
cells, thus, reducing monocyte adhesion to the endothelial cells [5]. In patients with acute lym-
phocytic leukemia, high-dose MTX (HDMTX) caused reversible neurotoxicity in the form of
white matter injury [6].

In a study examining the brainstem auditory system in children (2–12 years) with acute
lymphoid leukemia who received MTX treatment, 60% of individuals aged 5 years or less
showed an auditory deficit [7]. The brainstem auditory-evoked potential assessment, used to
evaluate ototoxicity in patients undergoing chemotherapy, revealed that 80% of those tested
exhibited some form of change and latency delay, with auditory impairment in the lower brain-
stem being the most common. In addition, the combination of MTX and other cancer drugs
for the treatment of both solid and hematological malignant tumors caused ototoxicity [8].

Avenanthramides (AVNs) are phenolic compounds originally extracted from oat grain
(*Avena sativa* L.) and have a molecular weight of approximately 300 g/mol. These polyphenols
exhibit a high antioxidant capacity and are highly abundant in human food. Several AVNs
have already been found to exist in oats; one of the most prevalent type is AVN-C, which has
the highest antioxidant activity [9]. AVN-C inhibited the proliferation of vascular smooth
muscle cells and increased the production of nitric oxide (NO), resulting in the prevention of
atherosclerosis [10]. We have previously demonstrated that AVN-C has a protective effect
against noise-induced hearing loss (NIHL) and drug-induced hearing loss (DIHL) [11].

The aims of this study were to investigate the effect of high-dose MTX on normal hearing,
provide insights into the possible mechanisms involved in MTX-induced hearing loss, and
evaluate the efficacy of the antioxidant AVN-C in the prevention of MTX toxicity.

## Materials and methods

### Animal care maintenance and drugs

C57Bl/6 background mice (with each group comprising 7 mice, 4 males and 3 females) aged
4–6 weeks were used in this study. In vivarium, standard conditions to shelter the mice were
followed, and adequate food and water were provided. This study was carried out in strict
accordance with the recommendations in the Guide for the Care and Use of Laboratory Ani-
mals of Chonnam National University. The protocol was approved by the Committee on the
Ethics of Animal Experiments of Chonnam National University (CNUHIACUC-20027). All
surgeries were performed under anesthesia and were made to minimize suffering. The drugs
used in this study were MTX (Cat. M9929; Sigma-Aldrich, St Louis, MO, USA), AVN-C (SL-
340; 4-105A NINT Innovation Center, 11421 Saskatchewan Dr Edmonton, AB, T6G2M9,
Canada), and folinic acid calcium salt (FA) (47612, Sigma, St Louis, MO, USA). The following
drug dosages were used in this study: MTX 4 mg/kg (high dose), AVN-C 10 mg/Kg, and/or
FA 7 mg/kg, once each day for 7 days. For *in vivo* experiments, the drug was administered
intraperitoneally. In this study, ketamine (100 mg/kg) and xylazine (10 mg/kg) were used as
anesthetics. The mice in these experiments were administered 0.3 cc of 0.9% NaCl intraperito-
neally 5 hours post HDMTX treatment, every day beginning on the first day of HDMTX treat-
ment and continuing for 1 week after the HDMTX treatment. No mortality was observed
among the treated mice, and the animals were in good health overall. For *in vitro* investiga-
tions with HEI-OC1 cells, 0.2 μM MTX was combined with AVN-C 1 μM and/or FA 3 μM.

## Methotrexate detection in mouse body fluids

The MTX used in this study was dissolved in 0.9% normal saline for both *in vitro* and *in vivo* applications. To detect MTX in the mouse serum and perilymph, MTX 4 mg/kg was administered to the subjects intraperitoneally (IP), and controls received normal saline. Under anesthesia, the whole blood was extracted directly from the mice hearts and collected in a sterile Eppendorf (EP) tube, which was then left undisturbed for 30 minutes to allow coagulation. After centrifuging at $1500 \times g$ for 10 minutes at 20˚C, the serum was removed quickly and analyzed using liquid chromatography-mass spectrometry (LC-MS/MS, AB SCIEX 4000 Q Trap mass spectrometer, Shimadzu LC 20A System) to detect MTX. The mice were anesthetized, and their heads were fixed to receive perilymph. The subcutaneous fat layer was dissected after skin incision, with gentle removal of the muscles to expose the tympanic bulla periosteum. By the incremental removal of bony fragments, the bulla was encapsulated before uncovering the round window niche that was then gently penetrated using a glass pipette to harvest the perilymph. Every fluid retrieved was analyzed using an LC-MS/MS to detect MTX. The MS conditions were as follows: Turbo Ion Spray, 500˚C, MRM scan form—positive mode, 5500 V, CG 20, GS1 50, and GS 60 spray voltage (MTX m/z 316.169/163.000, loperamide m/z 477.223/266.200). Gemini C18 3.0 μm columns (150 mm × 3.0 mm) fitted with Gemini C18 guard cartridge (4.0 mm × 2.0 mm), with the column oven at 40˚C and autosamplers at 4˚C, were set as the local conditions. Mobile phase was ACN:deionized water = 40:60 (V/V) with 0.1% formic acid and the flow rate set at 0.3 ml/min. In 0.9% normal saline and loperamide, standard stock solution of 1 mg/mL MTX was used.

## Auditory brainstem response for assessing animal hearing

One month after drug treatment, we measured the auditory brainstem response (ABR) from click to tone burst stimuli. The ABR in the left and right ears was evaluated; the body temperature was maintained with the help of a heat therapy pump (#TP700, MI, USA). All animals were anesthetized with ketamine (120 mg/kg) diluted with xylazine (10 mg/kg), which was administered intraperitoneally, and the mouse was place in an audiometric booth. The following ABR stimulus frequencies were tested: 8, 16, 24, and 32 kHz, as reported previously [12]. TDT's MF1 Multi-Field Magnetic Speaker was used to optimize the free field utilized to test the hearing range of mice, rats, and guinea pigs [13]. We tested the stimulus intensity levels at each frequency in decreasing order, i.e., from 90 dB to 20 dB of the visual ABR threshold. The stimulus level was calibrated at the ear opening using a custom-made probe tube microphone.

## Evaluation of wave I

The obtained ABR threshold was reduced in stimulus intensity by 20 dB for each animal to identify the lowest intensity at which an ABR wave I was detected. The ABR threshold and the wave I amplitude were determined by analyzing the stacked waveforms using the program R (version 4.0.4, Free Software Foundation's GNU General Public, Austria). The extensive ABR data obtained were stored on floppy disks for later offline analysis of the amplitude and latency of the ABR components. The amplitude of wave I was described as the difference in magnitude between the first positive peak and the next negative peak at 90 dB SPL.

## Scanning electron microscopy of outer hair cells

To shorten the time between death and fixation (typically 2 minutes) at room temperature (RT), the cochlea was rapidly dissected out of the mouse skull bone surgically (one animal at a time) after giving anesthesia to the mouse and a hole was made at the apex. The fixative

(500 μL), comprising 4% paraformaldehyde and 2.5% glutaraldehyde in 0.1 M sodium cacody-late buffer, was carefully perfused through the open round window, exiting through the hole created at the apical turn. The tissues were then post-fixed overnight at 4˚C on a rotating plat-form with the same buffer, rinsed three times with distilled water, and decalcified for 2 hours in 5% ethylenediaminetetraacetic acid (EDTA) in 100 mM Tris (pH 7.4). The cochlear coils were cut open and post-fixed at 4˚C in 1% osmium tetroxide for 2 hours. The samples were then dehydrated with 50% to absolute ethanol by sequential ethanol rinses, dried at the critical point, mounted on carbon tab support inserts, and sputter-coated with platinum. Imaging was performed using a scanning electron microscope (COXEM EM-30AX Plus, Republic of Korea) with a beam energy of 15 kV. The number of outer hair cells in cochleae was quantified. The cochleae were separated into apical, middle, and basal turns, and the hair cells in each turn were counted at a magnification of 500X. For each group, the number of hair cells in 100 μm cochlear turn length was averaged. If the bundle of stereocilia was missing, a hair cell was considered absent.

### Ribbon synapses and cochlear neuron integrity

After anesthetizing the mice, the cochleae were extracted from the mouse head skull, and a gaping tear was formed directly at the distal turn of the cochlea. Then, 4% paraformaldehyde was passed through the round window to the apex, with 4 hours of post-fixation at 4˚C, under gentle rotation. In addition to bone demineralization, the cochleae were placed in 0.12 mM EDTA for 1 hour at 4˚C. From each cochlea, three small pieces were cut (base, middle, and apex). The tissue samples were immersed in blocking buffer (donkey serum: 0.1% PBS-T, at 1:100 dilution) for 1 h at RT and were immediately incubated with primary antibodies at 4˚C overnight. After washing three times with 0.1% PBS-T (30 minutes, each wash), the samples were incubated in secondary antibodies for 4 hours at RT. The samples were then rinsed three times with 0.1% PBS-T for 30 minutes. The samples were then stained with phalloidin and DAPI for 3 minutes and rinsed once in PBS for 30 min. The samples were mounted on glass slides with vector protection solution and analyzed using an LSM 800 laser scanning micro-scope (Carl Zeiss Microscopy GmbH, Germany). The following major antibodies and titers were used: C-terminal binding protein 2 (CtBP2) (1:100, # 612044, BD Transduction Labora-tories™), myosin-7a (1:200, # 25–6791, Proteus), Anti-Neurofilament 200 (NF200) (1:200, #8135, Abcam), phalloidin (PL) (1:1,000, # A12379, Cell signaling), and DAPI (1:10,000, Invitrogen).

### Counting of presynaptic ribbons of inner hair cells

Cochlear turn lengths were determined for each study group. A high-resolution confocal microscope (LSM 800 laser scanning microscope) was used to generate confocal z-stacks of three regions from each cochlea. Image stacks were translated into image-processing software. Each cochlear turn (apex, middle, and base) contained at least 12 IHCs. We chose three visual areas of 20 μm each for each turn of the cochlea in the same way for all groups, demarcated them with a square, and counted presynaptic ribbons (CtBP2 punctates) of the IHCs that were found around IHCs as well as within nuclei as previously reported [14, 15]. We used seven mice from each study group to calculate the average number of presynaptic ribbons per IHC.

### HEI-OC1 cell culture

The House Ear Institute-Organ of Corti 1 (HEI-OC1) cells used in these experiments were a kind gift from Professor Hun Yi Park from Ajou university hospital, South Korea and were cultured under permissive conditions (33˚C). The media consisted of Dulbecco's Modified

Eagle Medium–high glucose (Gibco BRL, Gaithersburg, MD, USA) supplemented with 10% nonantibiotic fetal bovine serum (Gibco BRL), and 50 U/mL gamma interferon (Genzyme, Cambridge, MA, USA). AVN-C was diluted in DMSO and administered at a dose of 1 μM; MTX was dissolved in 0.9% normal saline and administered at a dose of 0.2 μM to induce cytotoxic effects on HEI-OC1. FA was dissolved in water and administered at a dose of 3 μM. AVN-C and FA treatment was provided 3 hours prior to the administration of MTX.

## MTT assay for cell viability assessment

Approximately 25 mg of 3-(4,5-dimethylthiazol-2-yl)-2,5-diphenyltetrazolium bromide (MTT; Sigma-Aldrich) was dissolved in 5 mL of PBS to obtain a reagent for this assay. A total of 20 μL of MTT reagent was added to each well, and the mixture was incubated at 37˚C for 30 minutes. Then, 100 μL of DMSO was added to each sample. The samples were incubated at 37˚C for 4 hours; the optical density was measured at 570 nm using a scanning electron spectrophotometer (Molecular Devices, SpectraMax ABS Plus #202–6262). The optical density of formazan in the solutions was measured using glass cuvettes for the spectrophotometer. All estimations within the plate reader were made by performing assays in 96-well plates, and the average optical density in control cells was assumed to be 100% and n = 7.

## Measurement of ROS

After drug treatment of HEI-OC cells, a Reactive Oxygen Species Detection Assay Kit was used to assess ROS levels in the cells. The cells were incubated at 37˚C in a humidified incubator with 5% $CO_2$ for 30 minutes before they were suspended. This kit uses the cell-permeable reagent 2′,7′-dichlorodihydrofluorescein diacetate (DCFDA), a fluorogenic dye that quantifies the activity of hydroxyl, peroxyl, and other ROS within the cell [16]. HEI-OC cells were rinsed once in 1× buffer and suspended in a microplate reader; fluorescence was recorded at a maximum excitation and emission wavelength of 495 nm and 529 nm, respectively, using a flow cytometer (BD FACSCalibur™, BD Biosciences, San Jose, CA, USA). Change in ROS levels was expressed as a percentage of control after background subtraction using Kaluza Analysis Software (Beckman Coulter, Inc., Brea, CA, USA).

## RNA isolation and real-time PCR

Treated HEI-OC1 cells were harvested, and total RNA was extracted using the TRIzol reagent (Invitrogen). The amount of RNA was determined using a full-spectrum spectrophotometer (NanoDrop ND-1000, Technologies Inc., Wilmington, DE, USA) with the absorbance measured at A260/A280 nm, and the results were analyzed using ND-1000 software. Complementary first strand DNA (cDNA) was synthesized from RNA using a reverse transcription cDNA synthesis kit (1st strand cDNA Synthesis kit; Takara PrimeScript™, Japan). Taq Master Mix was used for RT-PCR (Bioscience, Germany). The experiments were repeated thrice, in triplicates. The cycle conditions were as follows: denaturation occurred at 95˚C for 10 minutes and 10 seconds, whereas annealing occurred at 62˚C for 20 seconds followed by 72˚C for 30 seconds in 60 cycles. The expression levels were estimated using the $2^{-\Delta\Delta Ct}$ method, and the relative mRNA expression was normalized to glyceraldehyde 3-phosphate dehydrogenase expression (GAPDH). The primers used for each gene were as follows: GAPDH_For (5′–ACCACAGTCCATGCCATCAC–3′), GAPDH_Rev (5′–TCC ACC ACC CTG TTG CTG TA–3′), IL-1b_For (5′–GCTGCTTCCAAACCTTTGAC–3′), IL-b_Rev (5′–AGGCCACAGGTAT TTTGTCG–3′), IL-6_For (5′–TCCAGTTGCCTTCTTGGGAC–3′), IL-6_Rev (5′–GTACTCCAG AAGACCAGAGG–3′), TNFa_For (5′–CCACCACGCTCTTCTGTCTA–3′), TNFa_Rev (5′– CACTTGGTGGTTTGCTACGA–3′), BAX_For (5′–CTACAGGGTTTCATCCAG–3′), BAX_Rev

(5′–CCAGTTCATCTCCAATTCG–3′), HRK_For (5′–ATTCCGTACCTGTGCATGCCTG–3′), and HRK_Rev (5′–TGTGCTGAACAGTTGGTCCACG–3′).

### Data analysis

The data were analyzed using the Student's t test or one-way ANOVA with post-hoc Tukey–Kramer comparison tests. All statistical studies were performed via GraphPad Prism Software version 8.0. For p values of less than or equal to 0.05, the results were considered statistically significant. The number of repeats used for each experiment is described in the corresponding figure legends.

## Results

### MTX penetrates the blood-labyrinth barrier after systemic administration

We assessed for the presence of MTX in the cochlear fluid after systemic administration to determine whether it has a direct effect on the cochlea. To achieve this, mice were administered with 4 mg/kg of MTX intraperitoneally, and blood was collected at various time intervals (30 minutes and 1, 2, 3, 6, and 8 hours) and analyzed using LC-MS/MS to determine the level of MTX in the serum (Fig 1A). In wild-type (WT) mice, the blood serum level of MTX peaked at 1,490±2.8 ng/mL 30 minutes after IP treatment and rapidly declined thereafter (Fig 1B).

The presence of MTX in the perilymph was evaluated at several time points (1, 2, and 3 hours) after IP injection of MTX in WT mice to determine whether MTX passes the blood-labyrinth barrier (BLB). The levels of MTX in the perilymph peaked at 93.2±4.2 ng/mL 1 hour after systemic injection, demonstrating that MTX penetrated the BLB (Fig 1C). After 2 and 3 hours, the level of MTX was much lower in the perilymph.

### High-dose MTX causes considerable hearing loss *in vivo*, whereas AVN-C and FA protect hearing against MTX ototoxicity

Our *in vivo* study had shown that HDMTX causes an increase in the hearing thresholds in mice administered MTX. Subsequent assessments one month post treatment demonstrated a significant increase in the hearing thresholds for click sounds (***p<0.001; Fig 2C) in addition to the tone burst frequencies, at all tested frequencies (***p<0.001) in WT mice as compared to that noted in untreated control mice (Fig 2D).

Furthermore, we also examined the efficacy of AVN-C and FA against HDMTX-induced ototoxicity. Administration of AVN-C and FA 1 hour prior to MTX treatment markedly reduced the hearing thresholds for click sounds (21.4±2.4 dB and 28.9±3.9 dB SPL, respectively) in WT mice (Fig 2C) as compared to that in the group that received MTX alone. Moreover, the AVN-C and FA treatments also significantly reduced the hearing thresholds for tone bursts (***p<0.001) at all tested frequencies (Fig 2D) as compared to that in the control mice. Simultaneous administration of AVN-C and FA 30 minutes prior to MTX treatment resulted in a reduction of hearing thresholds for click sounds (23.6±2.4 dB SPL) (Fig 2C) and tone bursts (***p<0.001) at all tested frequencies as compared to that in the control mice (Fig 2D).

### AVN-C protects OHCs from MTX-induced ototoxicity

To characterize the protective effect of AVN-C on auditory hair cells against MTX-induced ototoxicity, we assessed for damage to the outer hair cell (OHC) stereocilia. Scanning electron microscopy (SEM) was performed to compare a normal cochlea with a treated cochlea 1 month after systemic administration of MTX, AVN-C, and FA. We observed that several OHCs were extinguished in the MTX-treated group compared with those in the other groups

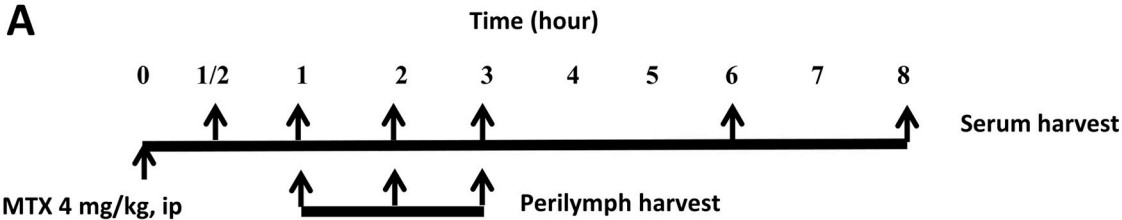

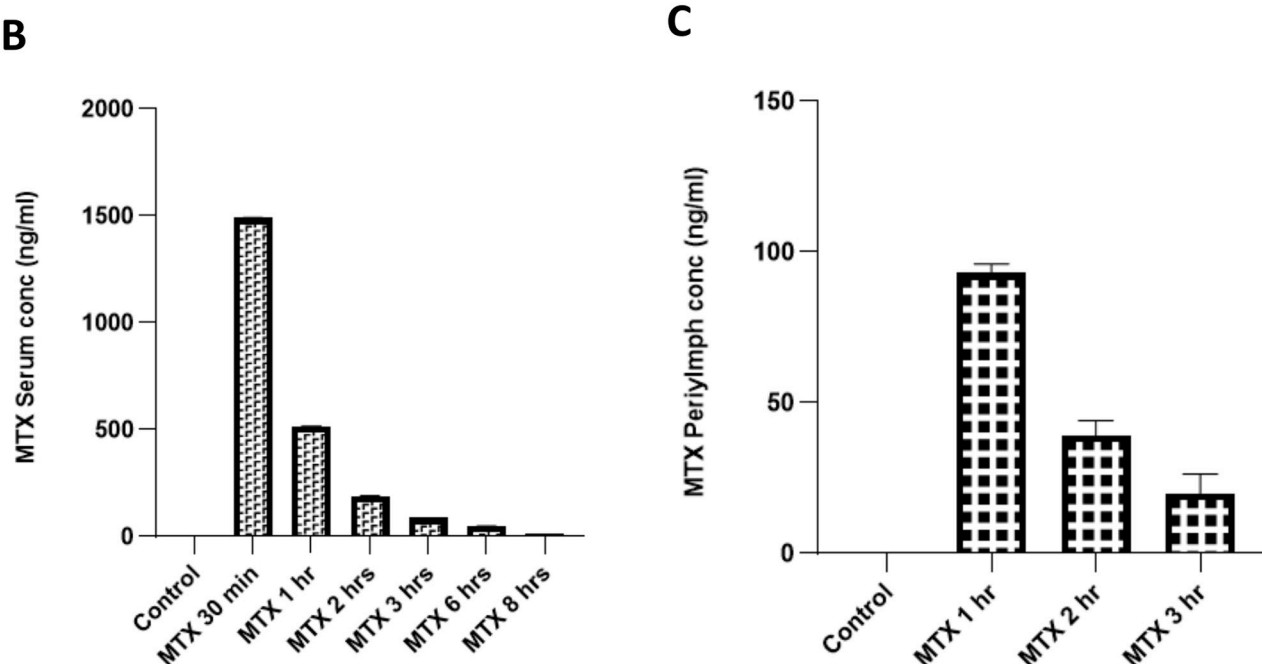

**Fig 1. Methotrexate detection in mouse body fluid by LC-MS.** A. Schematic diagram of time and treatment schedule. 0 hour represents the time of administration of 4 mg/kg of MTX to mice via an intraperitoneal injection; ½, 1, 2, 3, 6, and 8 hours stand for the time points of serum harvest; 9th hour indicates the time point for LC-MS analysis (upside). For perilymph, 1, 2, and 3 hours denote the moment perilymph was collected (below side); and 4th hour is the time point for LC-MS analysis to assess the presence of MTX in the perilymph. B. Results of serum analysis and illustrates the level of MTX in the serum being peaked after 30 minutes and then gradually declining, suggesting that MTX is taken up by different tissues. C. MTX was detected in the perilymph 1 hour after its administration in mice, and gradually decreased in the subsequent hours. Each group contained three mice (n = 3).

(Fig 3A), and damage to OHCs was noted mainly at the base (14±5.8 OHCs), middle (18.9±4.7 OHCs), and apex (20.4±3.8 OHCs) turns of the cochlea (Fig 3B–3D). In the FA+MTX-treated group, it was observed that OHCs were missing at the base (38±2.3), middle (39±4.6), and apex (40±7) turns (Fig 3A–3D). Meanwhile, AVN-C protected the OHCs at the base (49±2.4 OHCs), middle (49.3±1.9 OHCs), and apex (50.6±2.5 OHCs) turns (Fig 3A–3D). The combination treatment of AVN-C and FA limited the deleterious effect of MTX on OHCs to 42.3 ±3.7 OHCs at the base turn, 42.9±4.6 OHCs at the middle turn, and 43±4.1 at the apex turn (Fig 3A–3D). These OHCs are known to be affected by ototoxic drugs or noise. In the control group receiving the carrier, the number of OHCs was as follows: base turn, 43.7±2.4 OHCs; middle turn, 42.7±4.3 OHCs; and apex turn, 43.1±5 OHCs (Fig 3A–3D). The number of

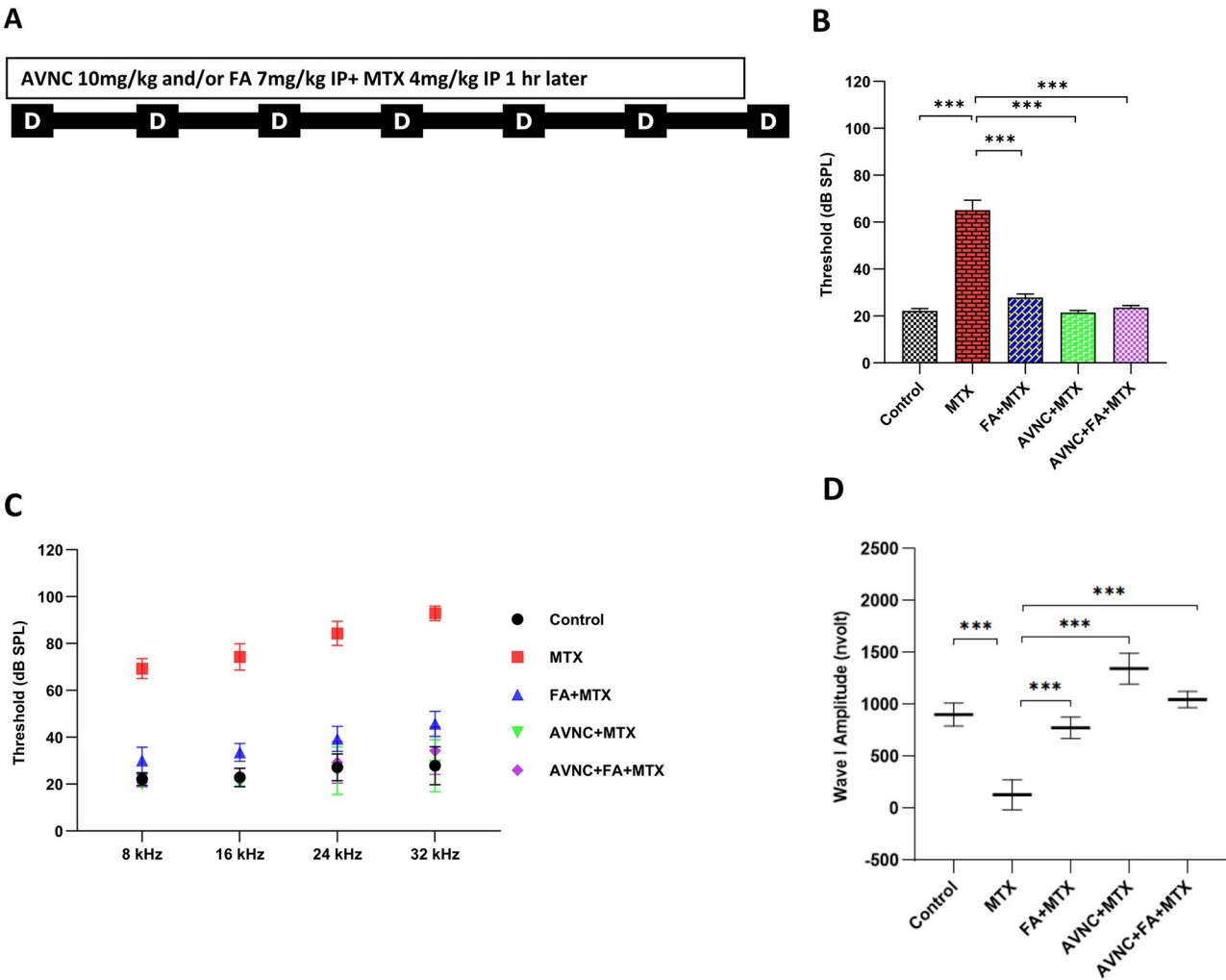

**Fig 2. AVN-C and FA preserve hearing from MTX ototoxicity by ABR.** A. Schematic diagram of drug treatment schedule and timeline. AVN-C and FA were injected intraperitoneally one hour before MTX administration for seven consecutive days. B. ABR wave I amplitude in 90 dB SPL. Wave I amplitude was defined as the magnitude difference between the first positive peak and the next negative peak. MTX treatment decreased the wave I amplitude, whereas treatment with AVN-C and FA reversed this decrease (***$p < 0.001$; one-way ANOVA). AVN-C treatment outperformed the rescue of wave I amplitude by FA as well as the combination treatment with AVN-C and FA. C (Click ABR) and D (Tone burst ABR) show that one month after drug treatment, the hearing thresholds in ABR increased in the MTX-treated group, whereas AVN-C and FA treatment reduced hearing thresholds (***$p < 0.001$; one-way ANOVA). n = 7 mice per group.

OHCs was counted per 100 μm, and a $p < 0.001$ was considered significant in all experimental groups. IHCs appeared normal and were well preserved in all groups.

## AVN-C treatment inhibits MTX-induced synaptic ribbon damage

The synaptic ribbon in the cochlear whole-mount preparations obtained from mice euthanized immediately after the ABR measurements were labelled with RIBEYE/CtBP2 and counted to directly test the following hypotheses: (1) AVN-C prevents synaptic ribbon loss and (2) MTX causes loss of synaptic ribbons of the cochlea IHC bands. Synaptic ribbons were counted in the cochlea regions corresponding to the tested stimulus frequencies that triggered ABR. Results showed that MTX damaged and reduced the number of synaptic ribbons of IHCs throughout the cochlea (Fig 4A). The CtBP2-positive signal count as determined by IHC

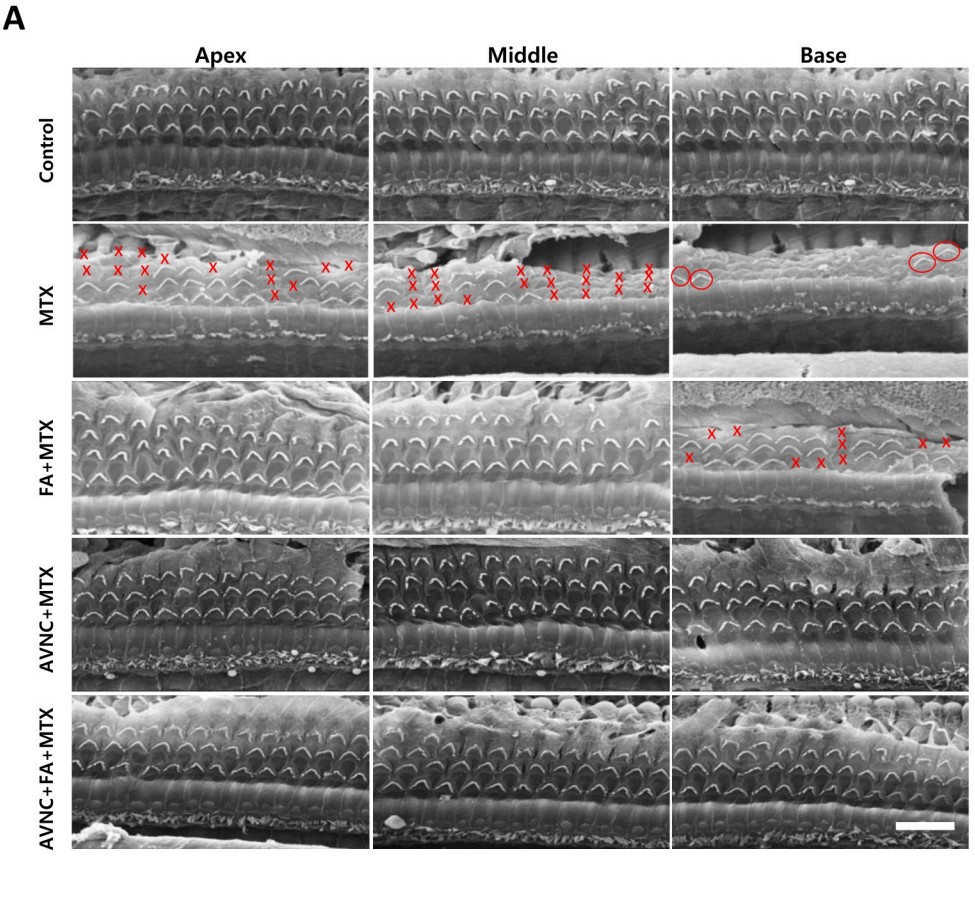

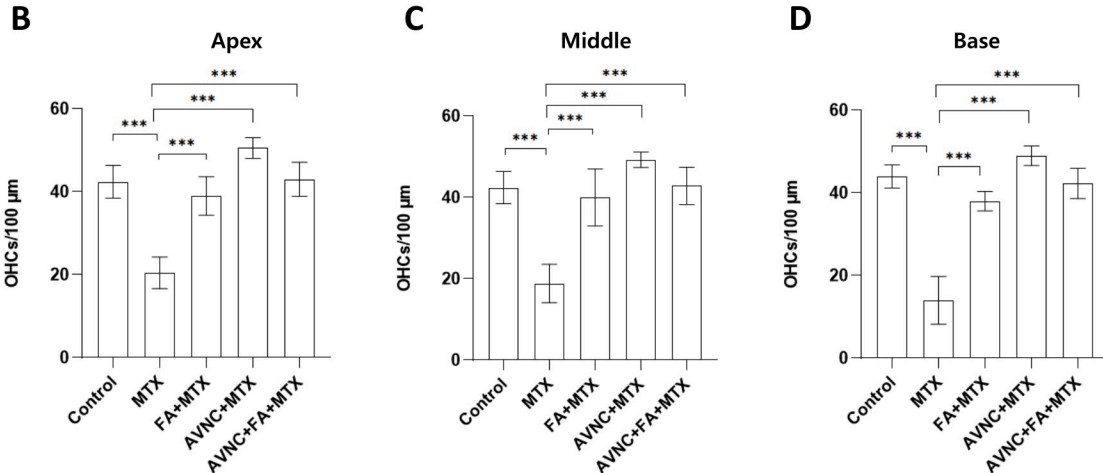

**Fig 3. AVN-C and FA MTX defend OHCs from MTX-induced ototoxicity by ABR.** A. In all cochlear turns, MTX severely damaged the OHCs. When administered 1 hour before MTX, AVN-C and FA shielded the OHCs against MTX-induced ototoxicity. B, C, and D. AVN-C treatment retained a greater number of OHCs at the cochlea turns than FA treatment. A, B, C, and D: ***p<0.001; one-way ANOVA. n = 7, and scale bar: 10 μm. X represents loss of OHC and O indicates the presence of OHC in the figure (red color).

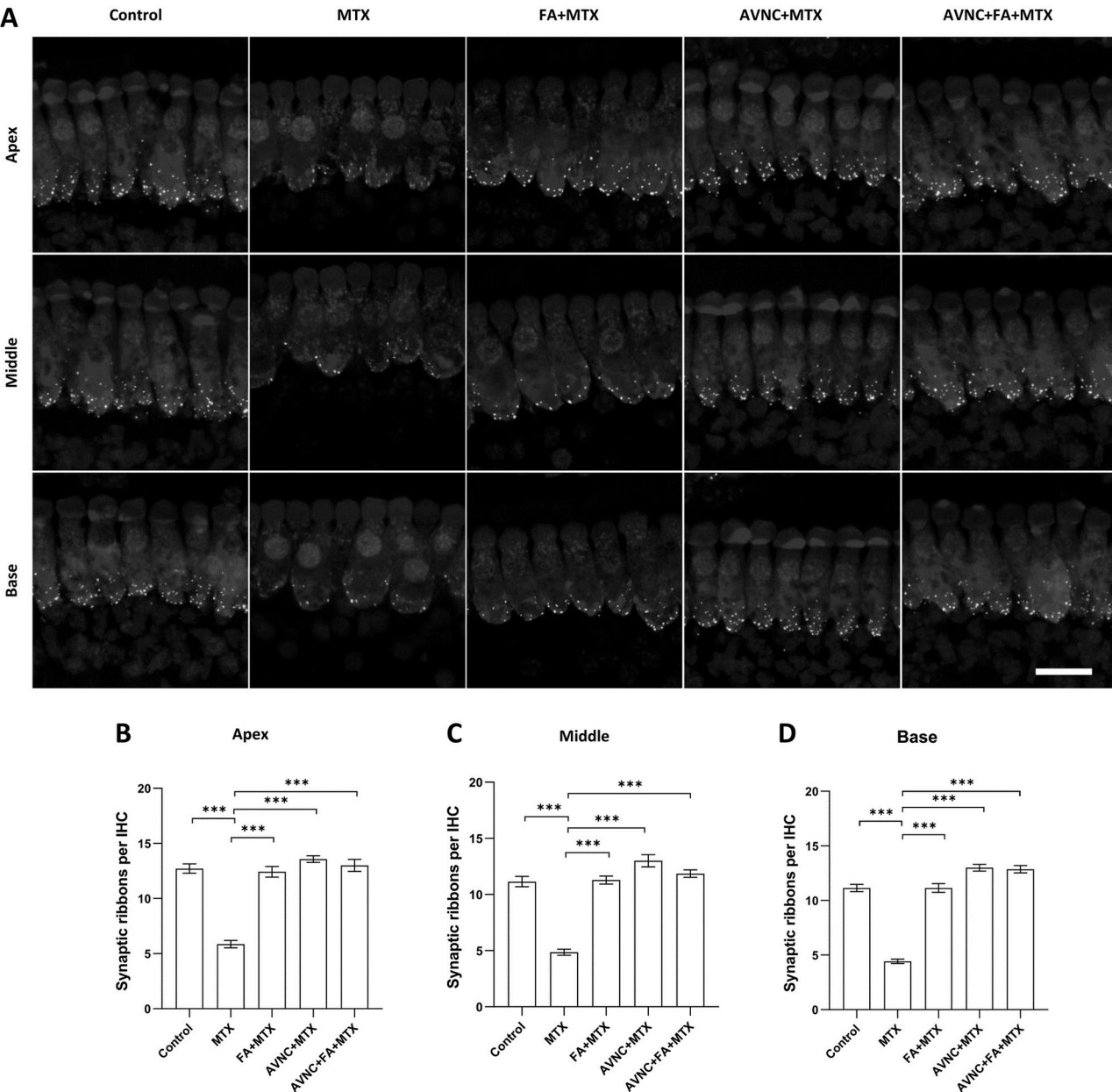

**Fig 4. AVN-C blocks synaptic ribbon damage caused by MTX treatment.** A. The number of CtBP2 positive signal (synaptic ribbons) in the MTX-treated group was significantly decreased in all three turns of the cochlea. B, C, and D. In the MTX-treated group, the number of synaptic ribbons in the base (4±0.7), middle (5±0.7), and apex turns (6±0.9) were reduced. The AVN-C-treated group recorded a high number of synaptic ribbons at the base (13±0.8), middle (13±1.4), and apex turns (14±0.8). The FA+MTX-treated (11±1.1; 11±1, and 12±1.3, respectively), AVN-C+FA+MTX-treated (13±1.3, 12±0.9, and 13±1.4, respectively), and control groups (13±1.1, 11±1.2, and 11±0.9, respectively) maintained a significant number of synaptic ribbons in the base, mid, and apex turns. ***p < 0.001. The Fig 4A is presented in black and white to clearly distinguish the distribution of cochlear synapses in IHCs of the cochlea. The white dots represent synaptic ribbons stained with CtBP2, whereas the inner hair cells were stained with myosin-7a. n = 7, and scale bar: 20 μm.

revealed that MTX treatment resulted in a significant decrease in the number of synaptic ribbons in all three turns of the cochlea among the group that received only MTX from the cochlea: apex (6±0.9), middle (5±0.7) and base (4±0.7) (Fig 4B–4D). On the other hand, the AVN-C-treated group maintained high numbers of these synaptic ribbons at the apex (14

±0.8), middle (13±1.4), and base (13±0.8) turns (Fig 4B–4D). The numbers of synaptic ribbons in the FA+MTX-treated (12±1.3, 11±1, and 11±1.1, respectively), AVN-C+FA+MTX-treated group (13±1.4, 12±0.9, and 13±1.3, respectively), and control groups (13±1.1, 11±1.2, and 11 ±0.9, respectively) at the apex, middle, and base turns and were significant (***$p < 0.001$) (Fig 4B–4D).

In addition, to assess the effect of MTX on hearing, we used the R program (version 4.0.4) to extract wave I at 90 dB SPL from the stored ABR raw data of each group of tested mice. The wave I amplitudes were severely decreased in the MTX-treated group (38.6±6.2 nanovolts), while it was higher in the remaining groups (control: 903.6±10.3, FA+MTX: 687.9±10.8, and AVN-C+MTX: 1,345.1±13.7 nanovolts). The combined treatment of AVN-C and FA 1 hour before MTX brought this wave I amplitude to 1,048.3±10.7 nanovolts (Fig 2B).

## AVN-C shelters the spiral ganglion neurons from axonal degeneration

To assess cochlear neurodegeneration in the turns, we stained all cochleae with anti-neurofilament 200 (NF200) and observed the cochlear morphology under a confocal microscope. Degeneration of cochlear neurons was more pronounced in the basal turn of the cochlea than in the middle and apex portions in the MTX-treated group. In the FA+MTX-treated group, the continuity and integrity of the cochlear neurons in the ascending position at the basal turn were disrupted, but those of the neurons in the middle and apex turns were intact. Notably, administration of the antioxidant AVN-C as well as the combination of AVN-C and FA resulted in a well-defined neuronal integrity of the cochlea, but MTX treatment caused tremendous neuronal degeneration throughout the cochlear turn (Fig 5B).

## AVN-C saves cells from MTX-induced cytotoxicity

The MTT assay, a known colorimetric assay that assesses both the metabolic activity of cells and the number of viable cells present, was performed on HEI-OC1 cells treated with MTX, AVN-C, and FA (alone and in combination with AVN-C). The viability of HEI-OC1 cells was evaluated in a concentration- and time-dependent manner. We initially determined the levels of MTX (Fig 6A) and AVN-C (Fig 6B) in HEI-OC1 cells by evaluating different dose regimens, before selecting the appropriate MTX and AVN-C doses to be used along with FA (Fig 6C). After completion of all treatment schedules and MTT assay, the absorbance of HEI-OC1 cells measured at 570 nm at 24 hours after MTX treatment showed that cell viability was as follow: control (100%), FA+MTX (66.2±13.1%), and AVN-C+MTX (86.3±8.2%). However, MTX significantly reduced the cell viability to 54.2±13.3%, resulting in cell death (Fig 6C).

## AVN-C decreases the ROS levels in MTX-induced HEI-OC1 cell ototoxicity

The cells were treated with cell-permeable DCFDA for 30 minutes at 37˚C in 5% $CO_2$, and after completing all treatment regimens cells were then processed for FACS analysis to determine ROS production. In HEI-OC1 cells treated with MTX alone, the fluorescein-DCFDA channel positive population produced considerably higher ROS levels than the control group (***$p<0.001$; Fig 7A). Three hours prior to the administration of MTX, treatment with AVN-C and FA—either alone or in combination—resulted in a significant reduction in ROS formation in HEI-OC1 cells (***$p<0.001$; Fig 7B).

Qualitative real-time polymerase chain reaction (RT-PCR) was performed to further dissect the downstream signaling pathways of AVN-C, FA, and MTX based on the changes in expression patterns of inflammatory cytokines. MTX upregulated all genes tested, including ROS- and apoptosis-related genes (TNFα, IL1β, IL6, BAX, and HRK); this was a common finding when experiment was repeated. The addition of AVN-C or FA appeared to significantly lower

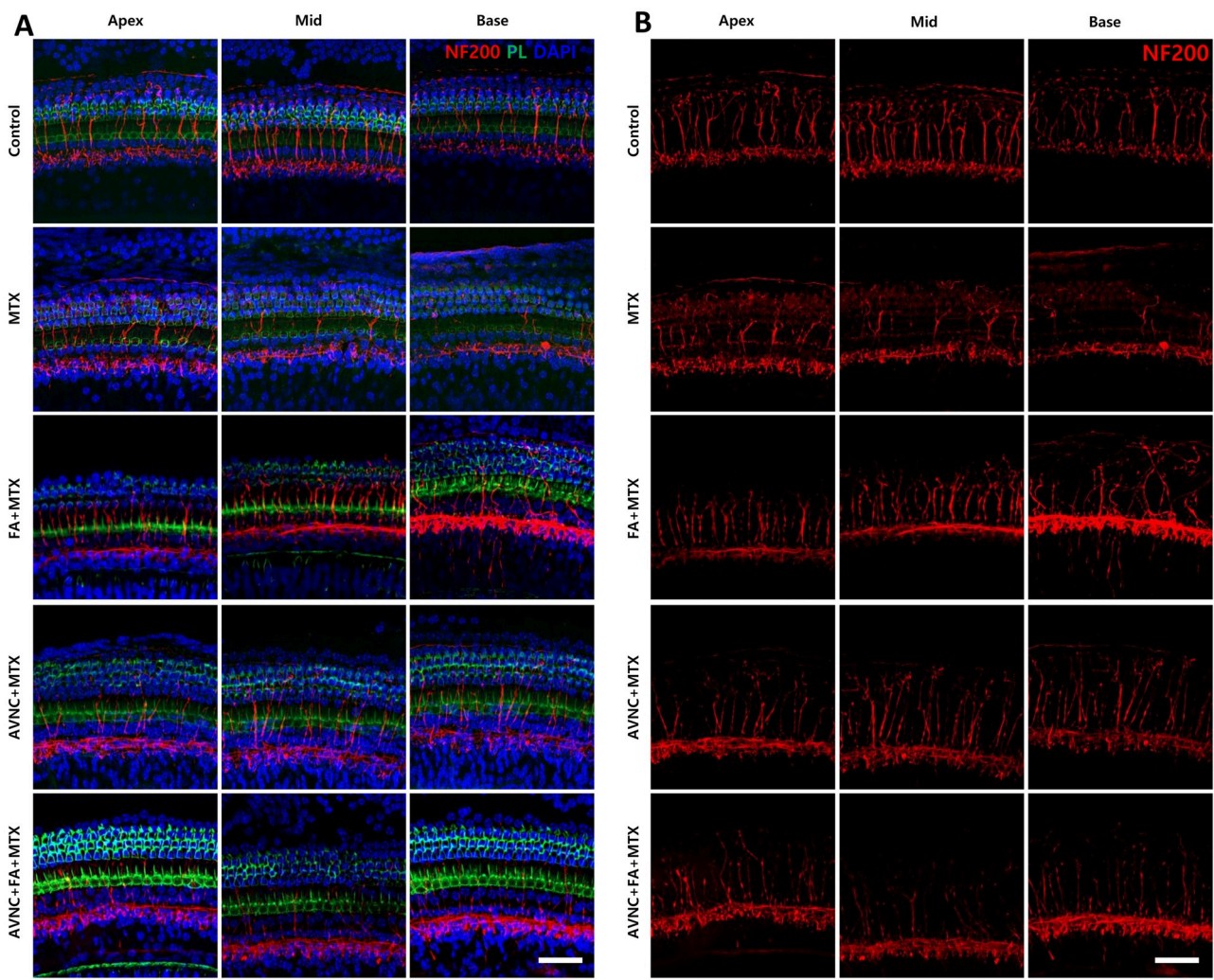

**Fig 5. AVN-C safeguards the cochlear neuron integrity from the detrimental effects of MTX.** Cochlear neurodegeneration was examined along cochlea turns with anti-neurofilament 200 (NF200) (Fig 5B). Under a confocal microscope, severe degeneration of cochlear neurons was observed, which was more pronounced at the basal turn than in the middle and apex portions in the MTX-treated group (Fig 5B). In the FA+MTX-treated group, the cochlear neurons at the basal turn showed discontinuity and deformity but preserved in the middle and apex turns of the cochlea. On the contrary, treatment with an antioxidant AVN-C alone as well as the combination of AVN-C and FA gave a well-defined cochlea neuronal integrity, while treatment with MTX disrupted cochlea neurons throughout the cochlear turns (Fig 5A and 5B). The following antibodies were used in staining: anti-neurofilament 200 (NF200), phalloidin (PL), and DAPI; the merged image shows the OHCs (Fig 5A): The scale bar used is 50 μm.

inflammation by decreasing the expression levels of these genes (Fig 8). The experiment was repeated seven times, and a p <0.001 was significant in all compared groups.

## Discussion

Although MTX-induced ototoxicity is not well studied, many studies have reported the occurrence of MTX cytotoxicity in other organs; an important underlying mechanism of MTX-induced toxicity is related to ROS production [17]. Hence, we evaluated the incidence of HDMTX-induced ototoxicity and assessed the preventive role of antioxidant AVN-C against this condition. The level of MTX in the bloodstream rapidly peaked 30 minutes after its administration in experimental mice, but decreased shortly thereafter, and was undetectable 8

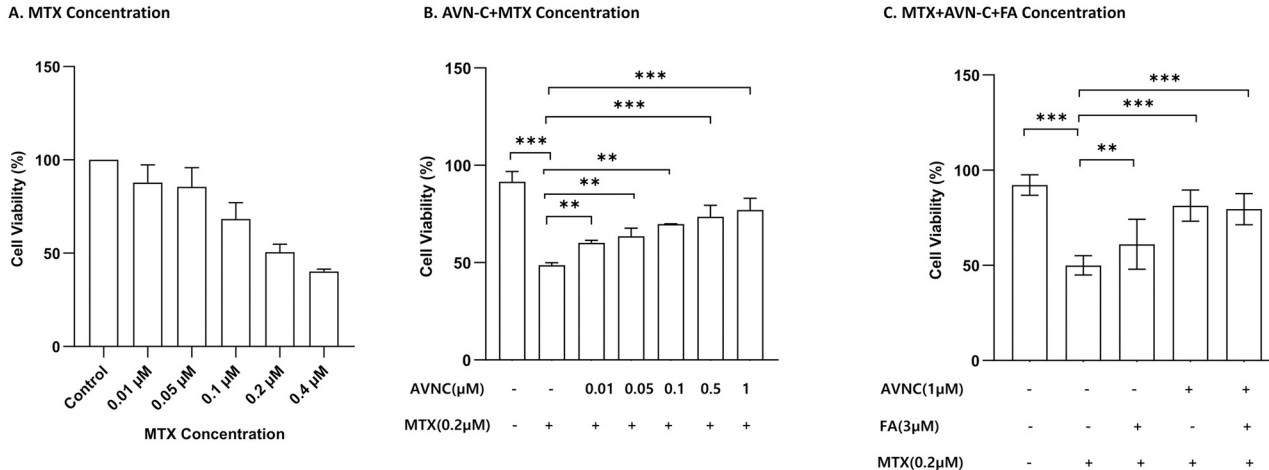

**Fig 6. MTX and AVN-C cell viability is dose-dependent.** Effect of MTX on the viability of HEI-OC1 cells. Panel A: Cell viability was significantly reduced at concentrations of MTX $\geq$ 0.2 µM. MTX reduced the cell viability in a concentration-dependent manner. Panel B: Pre-treatment of HEI-OC1 cells with various concentrations of AVN-C for 3 hours before addition of 0.2 µM of MTX for 24 hours provided the best protection. Panel C: HEI-OC1 cells were pretreated for 3 hours with 1 µM of AVN-C and 3 µM of FA simultaneously before being exposed to 0.2 µM of MTX for 24 hours. AVN-C alone or AVN-C concomitant to FA showed a significant protective effect against MTX cytotoxicity (***p<0.001), while FA showed moderate effect. ** p < 0.01; one-way ANOVA. Each group was tested seven times.

hours later (Fig 1B) owing to uptake of MTX in the plasma, spleen, liver, gastrointestinal tract, kidney, muscles, skin, and bone marrow [18]. The plasma levels of MTX have been evaluated as a function of time in Abcc3 knockout (KO) and WT mice [19]. These mice were all administered a single dose of MTX (10, 50, and 200 mg/kg) through intravenous (IV) bolus. The KO mice demonstrated considerably better total MTX clearance than the WT mice after 8 hours. This could explain the pharmacodynamics, absorption, and diffusion of MTX in many tissues. In addition, MTX is primarily distributed to the non-fatty tissues of the body after administration, and is rapidly transported across the capillary and cell membrane of the liver, kidney, and skin, allowing tissue to plasma concentration equilibrium ratios to be established on a time scale consistent with those of plasma flow limitation [18].

Meanwhile, we also measured the level of MTX in the perilymph after systemic treatment to check if it has a direct deleterious effect on the cochlea. The presence of MTX in the cochlea was confirmed by the appearance of MTX in the perilymph 1 hour after systemic injection (Fig 1C), which could explain the direct negative effect of MTX on the inner ear components. A previous study [20] had demonstrated that MTX administered at therapeutic doses can pass the blood-brain barrier and enter the cerebrospinal fluid. Following intravenous injection, less than 1 mM of MTX was found in the cerebrospinal fluid [21]. Similarly, platinum-based chemotherapeutic agents routinely used in oncology (namely cisplatin, carboplatin, nedaplatin, and oxaliplatin) have diverse ototoxic and neurotoxic effects. When the BLB is disrupted by treatment with diuretics or noise exposure, the uptake of drugs is increased and the extent of damage is greatly enhanced [22]. Our results showed that MTX crosses the BLB and causes direct damage to the OHCs. We had previously established the biodistribution and bioavailability of AVN-C in the body fluids of experimental mice; AVN-C was transited substantially longer in the perilymph than in the serum before it was washed out of the mouse cochlea [11].

A prospective open-label study on 11 patients with treatment-refractory autoimmune hearing loss was previously conducted to assess the efficacy of low-dose MTX as long-term treatment for autoimmune hearing loss. At the start of this study, an improvement in audiometric

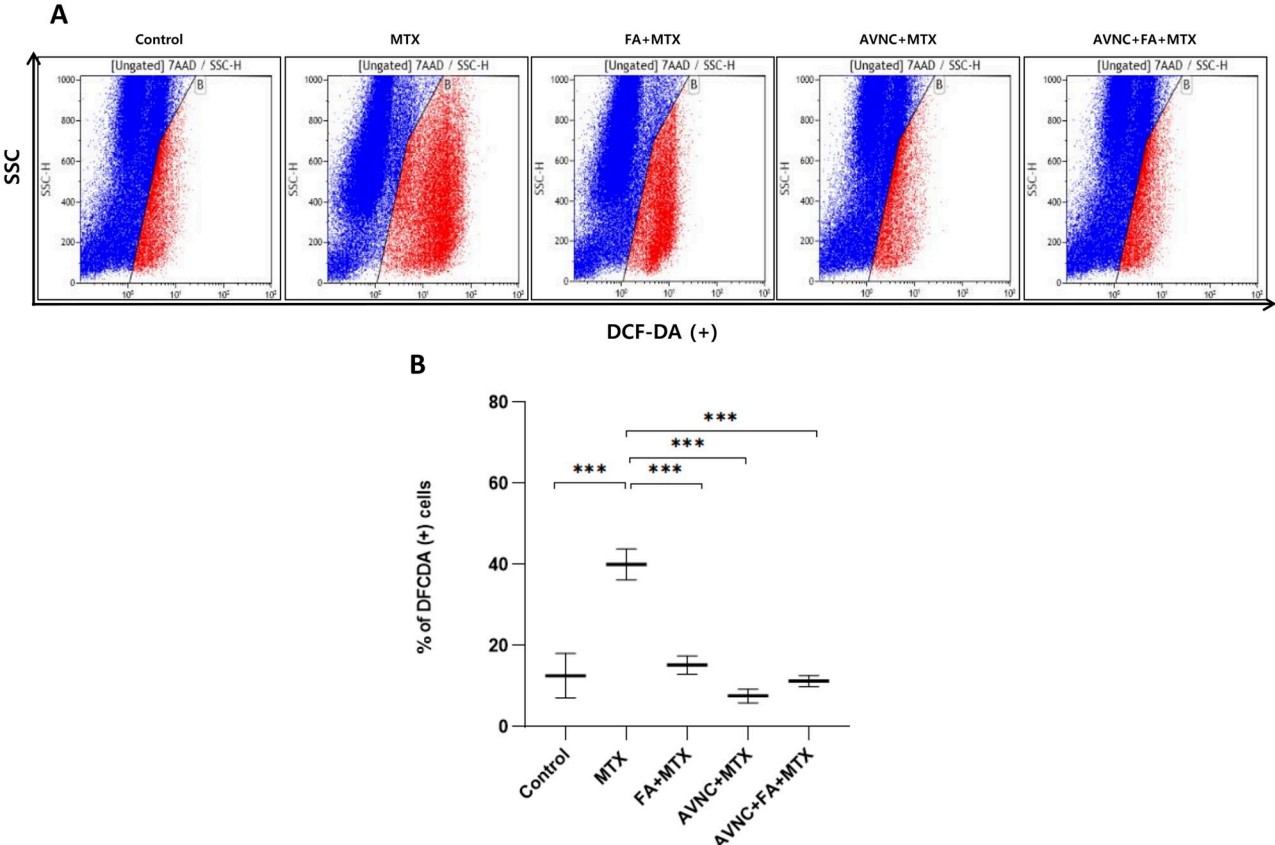

**Fig 7. AVN-C reduces the apoptotic effect caused by MTX treatment *in vitro*.** The obtained positive DCFDA percentages gated reactive oxygen species (ROS) after subtracting the background using the program Kaluza. All groups were scored seven times. The levels of intracellular ROS were determined by flow cytometry and DCFDA assay; the MTX-only-treated group produced high levels of ROS (Fig 7A). In the MTX-alone group, ROS generation increased, as demonstrated in the typical histograms of ROS fluorescence. On the other hand, pretreatment with AVN-C and FA alone or in combination, reduced ROS levels (Fig 7B). All groups tested had a significant p-value (*** p < 0.001; one-way ANOVA). Each group was tested seven times. After removing the background using the software Kaluza, the acquired positive DCFDA percentages gated ROS.

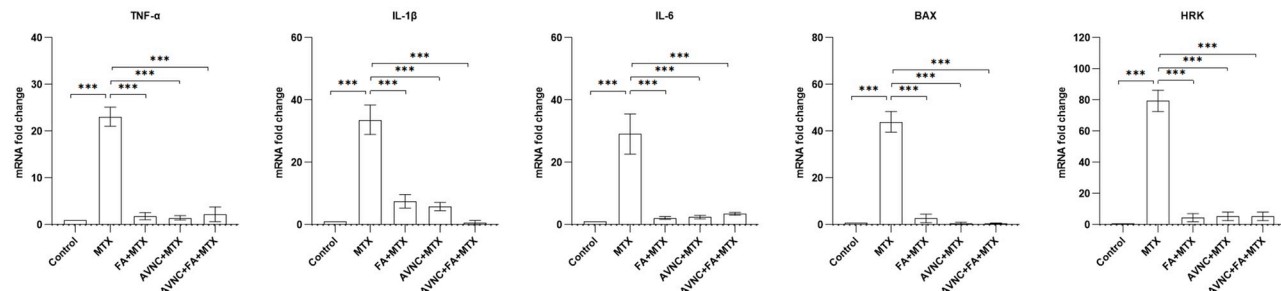

**Fig 8. AVN-C and FA downregulate the apoptotic genes in HEI-OC1 cells of MTX-induced ototoxicity.** The MTX ototoxicity model was established *in vitro* using HEI-OC1 cells. ROS- and apoptosis-related downstream genes such as TNF-a, IL1b, and IL6 were upregulated in the MTX-treated group. BAX, a death-inducing member, and HRK, an apoptosis activator, were highly upregulated in the MTX-treated group. However, either alone or in combination, AVN-C and FA significantly normalized the expression of these genes in HEI-OC1 cells. n = 3 (n represents triplicates) and ***p<0.001.

parameters (Permanent Threshold (PT) by >10 dB or standard deviation (SD) by >15%) was observed for at least one ear. Long-term treatment with low-dose MTX has been since proven effective in some patients with hearing loss that is thought to be mediated by autoimmunity and resistant to conventional therapies [23]. Another study showed that intratympanic injection of MTX does not cause any ototoxic effects; this study assumed that this method can be safely applied and used as a safe treatment alternative for autoimmune vestibulocochlear diseases [24].

In our investigation, HDMTX treatment caused DIHL with permanent threshold shifts—both in the click sound and in all evaluated frequencies of tone bursts—one month after treatment (Fig 2C and 2D). Methotrexate dosages are classified into three groups clinically. The first group is low-dose methotrexate, which is defined as less than 20 mg/m$^2$ (LDMTX $\leq$ 0.66 mg/kg), that is frequently used for rheumatological disorders such as rheumatoid arthritis. The second category includes intermediate IV doses ranging from 100–500 mg/m$^2$ (3.3–16.6 mg/kg) and are designated for breast and other solid tumors, as well as low-grade leukemias and lymphomas. The final category includes high-dose MTX, defined as a dose greater than or equal to 500 mg/m$^2$ (HDMTX $\geq$ 16.6 mg/kg), that is the mainstay IV chemotherapy for primary central nervous system lymphoma, osteosarcoma, and high-grade leukemias/lymphomas such as non-Hodgkin lymphoma; these treatment regimens are administered over several days [25–27]. Accordingly, we decided to administer a HDMTX treatment to WT mice for seven consecutive days in our present investigation as it is in line with the dosage employed in the clinical setting. Moreover, there is a lack of information in the literature about the death rate and general health concerns associated with high-dose MTX. In addition, several other clinical trials and follow-up studies of drugs, such as methotrexate, showed that they are beneficial in reducing morbidity measures, but their effect on mortality in people with rheumatoid arthritis remains uncertain [28]. Ambulatory high-dose methotrexate injection was used as a central nervous system prophylactic in patients with aggressive lymphoma and it was shown that MTX penetrates cell membranes, especially at doses high enough to cross the blood-brain barrier. MTX is heavily linked with albumin in the plasma circulation (50–80%), which explains the potential of severe toxicity associated with high-dose treatment in different organs. Patients whose MTX elimination is delayed are exposed to harmful MTX concentrations for an extended period, which can result in considerable morbidity. These toxins have the potential to cause long-term injury or death [29, 30]. LDMTX and HDMTX have different mechanisms, especially in cases such as renal failure, where many drugs can accumulate in the bloodstream [31]; this drug can cause treatment-related illness or even death. It is therefore particularly important to weigh the benefits against the risks.

The treatment protocols used in our investigations (AVN-C, FA, and AVN-C+MTX) all provided protection against HDMTX-induced toxicity, and AVN-C alone was efficient and overtopped FA alone or in combination, in respect of sheltering the synaptic ribbons and neuronal integrity (Figs 4A–4D, 5A and 5B). In young men and women, oat AVNs supplementation has been shown to reduce circulating inflammatory cytokines and inhibit the production of chemokines and cell adhesion molecules produced by downhill running [32]. Furthermore, administration of 10 mg/kg of AVN-C has been reported to prevent hearing loss in normal mice that were exposed to noise and ototoxic drugs (furosemide and kanamycin) [11]. Our findings pointed out that MTX caused oxidative stress that induced damage to the inner ear tissues. As evidenced by the findings of ABR and SEM, pre-administration of AVN-C to normal mice was effective in preventing or reversing the effects of MTX, with the AVN-C-treated group yielding outcomes like the control group that received the carrier (Fig 2C and 2D). Since the antioxidant AVN-C had protective effects against MTX toxicity, it can be assumed that MTX ototoxicity included the generation of ROS.

Folinic acid (FA) has been extensively examined and reported as an antidote of MTX, and it was employed in this study to assess the efficacy of AVN-C. It has been administered as an adjuvant treatment in studies with HDMTX [33–35]; moreover, we postulated that this could be the reason why hearing damage was not observed when treated with HDMTX. Subsequently, hearing impairment caused by oxidative stress in mice was prevented in NIHL and DIHL when the antioxidant AVN-C was administered [11]. However, combination treatment with AVN-C and FA before the administration of HDMTX had no additional effects, implying that AVN-C and FA share the same pathway.

Furthermore, inequalities in hearing loss between sexes have significant consequences for knowledge gaps in the translation from non-clinical to clinical settings [36]. We used 4 males and 3 females in our study to check for statistical differences between sexes as the difference in sex-based needs for non-clinical versus clinical research can limit a comprehensive knowledge of sex-based mechanistic variables. Some reports showed that women of all ages have superior hearing to men [37]. Such disparities may play a role in understanding and explaining clinically significant sex differences, and they are almost certainly essential for developing successful therapeutic treatment options. However, we found no variation in findings attributable to sex differences among the wild type mice studied during our investigations.

Ribbon synapses connect the IHCs to the spiral ganglion neurons (SGNs), which are the primary synaptic structures in the sound conduction pathway and play a key role in sound signal transmission [38]. Damage to the ribbon synapses hinders transmission of sound and conduction to the brain (where it is interpreted as sound), thereby increasing the hearing thresholds, and causing hearing loss. In addition, MTX treatment impairs both the central and peripheral nervous systems, with the potential for neurotoxicity in the central auditory nervous system [7]. Our results showed a decline in the number of synaptic ribbons of IHCs in the MTX-treated group. This finding suggests that HDMTX has a direct harmful effect on the synaptic ribbons. Additionally, MTX caused the death of several axons of the auditory nerve fibers, which reduced the neural output of the cochlea, and impaired the sensitivity and optimization of auditory nerve fibers. Besides, wave I amplitude was significantly decreased, and this implies a decrease in the firing of electrical impulses from the cochlea to the brain for interpretation of sound and ultimately leading to hearing loss (Fig 2B).

Earlier reports have demonstrated that cisplatin-induced ROS accumulation and aging reduced the number of ribbon synapses in IHCs, resulting in synaptopathy and OHC loss. ROS-induced deterioration in ribbon synapses may be a prelude to HC loss, according to these findings [39]. In mice, noise exposure caused significant reductions in ABR wave I amplitudes as well as the loss of cochlear ribbon synapses [40]. ABR amplitudes have been used successfully to alienate synaptopathy in listeners, considering that wave I depicts the synchronous firing of many auditory nerve fibers in the spiral ganglion cells [41].

Despite a significant decrease in ABR wave I amplitude readings in the MTX group owing to oxidative stress, AVN-C treatment substantially increased the wave I amplitude and maintained it higher and superseded FA and the concomitant treatment of AVN-C and FA (Fig 2B). The synaptic ribbons were retained and preserved under our treatment regimens (AVN-C, FA, and AVN-C+FA) (Fig 4A–4D). This is the reasonable because AVN-C, as a powerful antioxidant capable of reducing the ROS levels, protected the synaptic ribbons from the damaging effects of ROS. Meanwhile, FA is widely known as an antidote to MTX and indeed served as an antidote to MTX toxicity in this study. However, AVN-C was more effective and improved cochlear nerve fiber axon integrity than FA, since some innervation distortion was observed at the basal turn in FA-treated cochlea (Fig 5B). Following the damage caused by

MTX, AVN-C appeared to promote cochlear neuron survival, which may have boosted the firing of electrical impulses from the cochlea to the brain, explaining the improved ABR results when compared with those treated with MTX.

Previous works have demonstrated the functional recovery of regenerated synapses in treated animals using round-window delivery of NT3 protein by evaluating the suprathreshold amplitude of ABR wave I in response to tone pips in the damaged cochlear frequency regions [42]. Based on the current widely accepted theory of mammalian cochlear mechanics, the fluid in the cochlear scalae interacts with the elastic cochlear partition to produce transversely oscillating displacement waves that travel along the cochlear coil [43]. Previous studies have shown that exogenous neurotrophins directly delivered to the cochlear fluids enhance the survival of cochlear neurons after the HCs are damaged by ototoxic drugs [44]. The antioxidative role of AVN-C was observed in all *in vivo* studies, as expected from our earlier work [11], and FA was found to block MTX ototoxicity.

To better understand the role of ROS in the mechanism of MTX-induced apoptosis and the effects of AVN-C and FA, we pre-treated HEI-OC1 cells with 1 μM of AVN-C and 3 μM of FA for 3 hours before administering 0.2 μM of MTX. After 24 hours of MTX treatment, all the cells were examined under a fluorescence microscope. We observed that MTX could cause apoptotic morphological changes such as chromatin condensation, membrane blebbing and shrinkage, and apoptotic body formation (data not shown). DCF, a fluorescent probe commonly used to detect total ROS in cells, was also used. We observed that MTX-treated HEI-OC cells exhibited an increased DCF-positive population, suggesting the induction of ROS production by MTX (Fig 7A). ROS are known to be extremely dangerous, triggering oxidative stress through the oxidation of biomolecules and resulting in irreversible cellular damage and cell death [45–48]. The inflammatory cytokines (TNF-α, IL1b, IL6, BAX, and HRK) were significantly upregulated in HEI-OC1 cells treated with MTX (Fig 8). The overexpression of BAX in the MTX-treated cell group indicates that MTX induces ROS generation via a mitochondria-mediated pathway, leading to an increase in inflammation. Previous reports have shown that MTX causes potent mitochondrial disruption and apoptosis in HL-60 and Jurkat T cells through the production of ROS [17]. The apoptotic morphological changes were significantly reduced when the cells were pretreated with ROS scavenger AVN-C, and FA (Fig 7B). The number of healthy cells was much higher in the AVN-C-treated group than in the FA-treated group (Fig 7A and 7B). Despite the use of FA (a well-known antidote for MTX toxicity), either alone or in conjunction with AVN-C, the antioxidant AVN-C was found to be significantly beneficial in lowering ROS production and inflammation, as well as avoiding ototoxicity in HEI-OC1 cells. AVN-C was previously found to protect HEI-OC1 cells from oxidative stress caused by gentamicin treatment [11]. Here, we demonstrated a way to protect these cells from MTX ototoxicity.

## Conclusion

First, we demonstrated that treatment with MTX causes significant hearing loss. When administered in large doses, it can penetrate the BLB, causing damage to the synaptic ribbons, cochlear neurons, and OHCs. The antioxidant AVN-C has a remarkable protective effect against HDMTX-induced ototoxicity, and we used FA to weigh the effect of AVN-C during our investigations. We showed that AVN-C protects the ribbon synapses, cochlear neuron integrity, and OHCs from the harmful effects of MTX and that it improves ABR. These findings suggest that AVN-C can be utilized for hearing preservation. Our findings emphasize that AVN-C is effective and can protect against the harmful effects of HDMTX and suggest that ROS is involved in the occurrence of MTX-induced ototoxicity.

## Supporting information

**S1 Fig. Methotrexate detection in mouse body fluid by LC-MS.**
(XLSX)

**S2 Fig. Wave I ABR amplitude.**
(XLSX)

**S3 Fig. ABR click and tone burst.**
(XLSX)

**S4 Fig. OHCs counting.**
(XLSX)

**S5 Fig. Ribbon counting.**
(XLSX)

**S6 Fig. MTT MTX HEI-OC1 cell viability.**
(XLSX)

**S7 Fig. AVNC+FA+MTX HEI-OC1 DCFDA.**
(XLSX)

**S8 Fig. qPCR data.**
(XLSX)

## Author Contributions

**Conceptualization:** Hyong-Ho Cho.

**Data curation:** Alphonse Umugire, Youngmi Choi, Sungsu Lee, Hyong-Ho Cho.

**Formal analysis:** Alphonse Umugire, Youngmi Choi, Sungsu Lee.

**Funding acquisition:** Hyong-Ho Cho.

**Investigation:** Alphonse Umugire, Youngmi Choi, Sungsu Lee, Hyong-Ho Cho.

**Methodology:** Alphonse Umugire, Youngmi Choi, Sungsu Lee.

**Resources:** Sungsu Lee, Hyong-Ho Cho.

**Software:** Alphonse Umugire.

**Supervision:** Hyong-Ho Cho.

**Validation:** Sungsu Lee, Hyong-Ho Cho.

**Visualization:** Alphonse Umugire, Youngmi Choi.

**Writing – original draft:** Alphonse Umugire.

**Writing – review & editing:** Sungsu Lee, Hyong-Ho Cho.

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
