## [Decision Letter · Decision Letter 0]

12 Oct 2021

PONE-D-21-29604Efficiency of Antioxidant Avenanthramide-C on High-Dose Methotrexate-Induced Ototoxicity in MicePLOS ONE

Dear Dr. Cho,

Thank you for submitting your manuscript to PLOS ONE. After careful consideration, we feel that it has merit but does not fully meet PLOS ONE’s publication criteria as it currently stands. Therefore, we invite you to submit a revised version of the manuscript that addresses the points raised during the review process.

We look forward to receiving your revised manuscript.

Kind regards,

Susan E Shore, Ph.D.

Academic Editor

PLOS ONE

Journal Requirements:

"This research was supported by a grant (NRF-2020R1A2C1007473) from the Basic Science Research Program through the National Research Foundation of Korea, funded by the Ministry of Education, Science and Technology. It was also supported by a grant from the Chonnam National University Hospital Biomedical Research Institute (BCRI20038)."

"This research was supported by a grant (NRF-2020R1A2C1007473) from the Basic Science Research Program through the National Research Foundation of Korea, funded by the Ministry of Education, Science and Technology. It was also supported by a grant from the Chonnam National University Hospital Biomedical Research Institute (BCRI20038)."

"This research was supported by a grant (NRF-2020R1A2C1007473) from the Basic Science Research Program through the National Research Foundation of Korea, funded by the Ministry of Education, Science and Technology. It was also supported by a grant from the Chonnam National University Hospital Biomedical Research Institute (BCRI20038)."

Additional Editor Comments:

Reviewer one has made detailed comments of your manuscript. I have reviewed the paper and concur with the reviewer. Please make all changes as requested and resubmit.

Reviewers' comments:

Reviewer's Responses to Questions

**Comments to the Author**

1. Is the manuscript technically sound, and do the data support the conclusions?

Reviewer #1: Partly

2. Has the statistical analysis been performed appropriately and rigorously? 

Reviewer #1: No

3. Have the authors made all data underlying the findings in their manuscript fully available?

Reviewer #1: Yes

4. Is the manuscript presented in an intelligible fashion and written in standard English?

Reviewer #1: No

5. Review Comments to the Author

Reviewer #1: This manuscript reported that a high dose of methotrexate (MTX), a chemotherapeutic agent, causes ototoxicity in mice that is associated with oxidative stress and pretreatment with the antioxidant avenanthramide (Avn-C) prevents MTX-induced loss of inner hair cell synaptic ribbons, loss of hair cells, and hearing loss. Next, HEI-OC 1 cells were employed to assess ROS and inflammatory response genes to confirm the effects of Avn-C. Additionally, the concentrations of MTX in the endolymph and serum were determined by HPLC. However, this manuscript is lacking in detail and needs significant revision. There are also many inconsistences in the manuscript. The authors should consider working with a professional scientific writer and editor to revise this manuscript.

Major Comments:

1. In the results section, the findings are not well described. For example, there are only two short sentences describing the findings for high-dose-MTX-induced hearing loss in vivo. There is no information on general health, death rate, or consideration for additional nutrition for supplementation after high doses of MTX treatment.

2. It is also confusing to have all figure legends inserted into the results section.

3. There is no paragraph in the methods section on how Avn-C was administered in this study, such as dose and route. In the animal section, it stats “all surgeries were performed under anesthesia”, but it is not known what surgeries were done and for which experiments.

4. The statistical data analysis section is over-simplified and confused. For example, it is mentioned “each experiment was performed seven times”, but RT-PCR experiments were repeated three times (page 14, line 231) and in the Fig.1 legend it describes “Panel C: Each group contained three mice (n = 3).” Also how were RT-PCR data analyzed?

5. The descriptions in the figure legends do not match the figures. For example, Fig. 2 A illustrates a schematic diagram of treatment, but in the legend ABR results were described in panel A. Fig. 2B showed wave I amplitudes, but there was no description of wave I amplitude in the legend.

6. There is a lack of discussion on how the high dose of MTX used in this study is relevant to clinical use for chemotherapy.

7. Some literature citations are not accurate. For example, page 26, lines 561–564, the citation of reference (32) was not associated with the supplementation of Avn-C at a dose of 0.1 g/kg in the diet of rats.

6. PLOS authors have the option to publish the peer review history of their article (what does this mean?). If published, this will include your full peer review and any attached files.

Reviewer #1: No

---

## [Author Response · Author response to Decision Letter 0]

2 Jan 2022

Efficiency of Antioxidant Avenanthramide-C on High-Dose Methotrexate-Induced Ototoxicity in Mice

PONE-D-21-29604

Thank you for giving us the opportunity to submit a revised draft of our manuscript titled “Efficiency of Antioxidant Avenanthramide-C on High-Dose Methotrexate-Induced Ototoxicity in Mice” to PLOS ONE. We truly appreciate the time and effort of the reviewers for providing their valuable feedbacks on our manuscript. We have been able to incorporate relevant changes to reflect most of the suggestions provided by the reviewers. The changes have been highlighted within the manuscript.

Here is a point-by-point response to the reviewers’ comments and concerns.

Comment:

Q1-1. In the results section, the findings are not well described. For example, there are only two short sentences describing the findings for high-dose-MTX-induced hearing loss in vivo. 

Response:

The wild-type (WT) mice used in this study were all administered high doses of MTX (HDMTX), and our results showed that these mice demonstrated significant irreversible hearing loss. We had described this finding briefly, as shown in the lines 287–290 of the originally submitted manuscript. However, the description of the subsequent results stretches into the following paragraph, which we have now combined and incorporated (pages 10–11, lines 288–304) in the revised manuscript. 

1. High-dose MTX causes considerable hearing loss in vivo, whereas AVN-C and FA protect hearing against MTX ototoxicity.

Our in vivo study had shown that HDMTX causes an increase in the hearing thresholds of mice administered MTX alone. Subsequent assessments one month post MTX treatment demonstrated a significant increase in the hearing thresholds for click sounds (p-value p***<0.001; Fig. 2C) in addition to the tone burst frequencies at all tested frequencies (p-value ***<0.001) in WT mice as compared to that noted in untreated control mice (Fig. 2D).

Furthermore, we examined the in vivo efficacy of AVN-C and FA against HDMTX–induced ototoxicity. Our study revealed that the administration of AVN-C and FA one hour prior to MTX treatment markedly reduces the hearing thresholds for click sounds (21.4±2.4 dB and 28.9±3.9 dB SPL, respectively) in WT mice (Fig. 2C) as compared to that in the group that received MTX alone. Moreover, the AVN-C and FA treatments significantly reduce the hearing thresholds for tone bursts (***p < 0.001) at all tested frequencies (Fig. 2D) as compared to that in the control mice. Furthermore, simultaneous administration of AVN-C and FA 30 minutes prior to MTX treatment was found to result in a reduction of hearing thresholds to 23.6±2.4 dB SPL for click sounds (Fig. 2C) and for tone bursts (***p < 0.001) at all tested frequencies as compared to that in the control mice (Fig. 2D). 

In addition, the histology findings described in the following paragraphs enhanced the degree of damage caused by HDMTX (pages 11–13, lines 306–364).

Comment:

Q1-2. There is no information on general health, death rate, or consideration for additional nutrition for supplementation after high doses of MTX treatment.

Response:

We have included the animal care protocol and general health condition after administration of HD-MTX in the methodology section as follows:

The mice in these experiments were administered 0.3 cc of 0.9% NaCl intraperitoneally five hours post HD-MTX treatment every day beginning on the first day of HDMTX treatment and continuing for one week after the HDMTX treatment. No mortality was observed among the treated mice, and the animals were in good health overall (page 4, lines 104–107). 

There is a lack of information in the existing literature regarding the death rate and general health concerns associated with high-dose MTX (page 16, lines 456–457). Folinic acid (FA) has been extensively assessed and reported as an antidote for MTX overdose, and it was employed in the present study to assess the efficacy of AVN-C. Various studies involving treatment with HDMTX have used FA as an adjuvant (31-33). Hence, we hypothesized that FA administration might be the reason why hearing damage was not observed in the studies with HDMTX treatment (page 17, lines 485–488). 

All this information has been incorporated in the revised manuscript. We are thankful to the reviewer for the insightful comment that has prompted us to improving the readability of our manuscript.

Comment:

2. It is also confusing to have all figure legends inserted into the results section.

Response:

PLOS One’s policy regarding figure captions states that figure captions should be placed in the manuscript text in the reading order immediately following the paragraph where the figure is first cited. It further says that captions should not be included as part of the figure files or submitted as a separate document; hence, we have complied with these manuscript formatting guidelines in the original manuscript. However, according to the reviewer’s suggestion, we have removed all the figure legends from the results section to the end of the revised manuscript to avoid ambiguity and promote better clarity.

Comment:

3. There is no paragraph in the methods section on how Avn-C was administered in this study, such as dose and route. In the animal section, it states “all surgeries were performed under anesthesia”, but it is not known what surgeries were done and for which experiments.

Response:

We are thankful to the reviewer for the useful suggestions. Accordingly, we have ensured much transparency regarding the drug treatment, dose, route of administration used in our study in addition to the details of the surgeries performed under anesthesia in our revised manuscript. This information now stands corrected and incorporated accordingly in the revised manuscript (page 4, lines 101-109). During this study, surgeries were performed to facilitate blood collection directly from the hearts of the mice, which is explained in detail (page 4, lines 115-117) in the revised manuscript. In addition, perilymph harvested through a round window surgical approach and the procedure used is detailed (page 5, lines 120-124) in the revised manuscript. The cochlea was extracted from mouse head skull surgically (page 6, lines 159-161, 174-175) and prepared for further analysis. All these data are included in the revised manuscript as follows.

The following drug dosages were used in this study: MTX 4 mg/kg (high-dose), AVN-C 10 mg/kg, and/or FA 7 mg/kg once each day for 7 days. For in vivo experiments, the drug was administered intraperitoneally. In this study, Ketamine 100 mg/kg and Xylazine 10 mg/kg were used as anesthetics. The mice in these experiments were given 0.3 cc of 0.9% NaCl intraperitoneally 5 hours after HDMTX treatment, every day beginning on the first day of HDMTX treatment and continuing for one week post HDMTX treatment. No mortality was observed among the treated mice and the animals were in good health overall. For in vitro investigations with HEI-OC1, 0.2 µM MTX was combined with AVN-C 1 µM and/or FA 3 µM (page 4, lines 101-109).

Under anesthesia, the whole blood was extracted directly from the mice hearts and collected in a sterile Eppendorf (EP) tube, which was then left undisturbed for 30 minutes to allow coagulation. (page 4, lines 115-117).

The mice were anesthetized, and their heads were fixed to receive perilymph. The subcutaneous fat layer was dissected after skin incision, with gentle removal of the muscles to expose the tympanic bulla periosteum. By the incremental removal of bony fragments, the bulla was encapsulated before uncovering the round window niche that was then gently penetrated using a glass pipette to harvest the perilymph. (page 5, lines 120-124)

To shorten the time between death and fixation (typically 2 minutes) at room temperature (RT), the cochlea was rapidly dissected out of the mouse skull bone surgically (one animal at a time) after giving anesthesia to the mouse and a hole was made at the apex (page 6, lines 159-161).

After anesthetizing the mice, the cochleae were extracted from the mouse head skull, and a gaping tear was formed directly at the distal turn of the cochlea (page 6, lines 175-176).

Comment:

4. The statistical data analysis section is over-simplified and confused. For example, it is mentioned “each experiment was performed seven times”, but RT-PCR experiments were repeated three times (page 14, line 231) and in the Fig.1 legend it describes “Panel C: Each group contained three mice (n = 3).” Also how were RT-PCR data analyzed?

Response:

The statistical analysis has been explained in further details (page 9, lines 262–267) in the revised manuscript. We have used seven replicates during the in vivo investigations and three repeats in the in vitro studies. We have described n = 3, where n represents triplicates for in vitro RT-PCR experiments (page 9, lines 247) and RT-PCR analysis is described in detail (page 9, lines 250-251, 262-267). Fig. 1 represents the detection of MTX in both the mouse body fluids, i.e. serum and perilymph from three mice (n = 3), as the used sample size was statistically adequate for the detection of MTX in mouse body fluids. 

The expression levels were estimated using the 2-ΔΔCt method, and the relative mRNA expression was normalized to glyceraldehyde 3-phosphate dehydrogenase expression (GAPDH). (page9, lines 250-251).

The data were analyzed using the Student's t test or one-way ANOVA with post-hoc Tukey–Kramer comparison tests. All statistical studies were performed via GraphPad Prism Software version 8.0. For p values of less than or equal to 0.05, the results were considered statistically significant. The number of repeats used for each experiment is described in the corresponding figure legends (page 9, lines 262–267).

Comment:

5. The descriptions in the figure legends do not match the figures. For example, Fig. 2 A illustrates a schematic diagram of treatment, but in the legend ABR results were described in panel A. Fig. 2B showed wave I amplitudes, but there was no description of wave I amplitude in the legend.

Response:

We thank the reviewer for their critical and helpful review of our manuscript. As per the reviewer’s concerns, the description of Fig. 2 in the figure legends has been rectified and rewritten in accordance with the figure panels, and the ambiguity has been addressed appropriately (page 23, lines 723–733). The description of wave I can be found in the paragraph marked "Evaluation of Wave I." In this paragraph, we have provided a detailed description of wave I and specified that we have evaluated it at 90 dB SPL (pages 5-6, lines 147-155). All these discussions have been incorporated in the revised manuscript.

Fig 2. AVN-C and FA preserve hearing from MTX ototoxicity by ABR. A. Schematic diagram of drug treatment schedule and timeline. AVN-C and FA were injected IP one hour before MTX administration for seven consecutive days. B. ABR wave I amplitude in 90 dB SPL. Wave I amplitude was defined as the magnitude difference between the first positive peak and the next negative peak. MTX treatment decreased the wave I amplitude, whereas treatment with AVN-C and FA rescued the decrease in wave I amplitude due to MTX treatment (*** p < 0.001; one-way ANOVA). AVN-C treatment outperformed the rescue of the decrease in wave I amplitude by FA as well as the combination treatment with AVN-C and FA. C (Click ABR) and D (Tone burst ABR) show that one month after drug treatment, the hearing thresholds in ABR increased in the MTX-treated group, whereas AVN-C and FA treatment reduced hearing thresholds (***p < 0.001; one-way ANOVA). n=7 mice per group.

Comment:

6. There is a lack of discussion on how the high dose of MTX used in this study is relevant to clinical use for chemotherapy.

Response:

The protocol of HDMTX treatment used in our study involved the administration of 28 mg/kg MTX per week i.e. 4 mg/kg once a day for seven consecutive days. Various reports corroborate the relevance of this dose to the applicable dose in the clinical settings (pages 15–16, lines 445–453).

Methotrexate dosages are classified into three groups in the clinic: the first group is low-dose methotrexate, defined as less than 20 mg/m2 (LDMTX ≤ 0.66 mg/kg), that is frequently used for treatment of rheumatological disorders such as rheumatoid arthritis. The second category includes intermediate intravenous (IV) doses ranging from 100–500 mg/m2 (3.3–16.6 mg/kg) and are designated for breast and other solid tumors as well as low-grade leukemias and lymphomas. The final category includes high-dose MTX, defined as a dose greater than or equal to 500 mg/m2 (HDMTX ⩾16.6 mg/Kg), that is the mainstay IV chemotherapy for primary central nervous system lymphoma, osteosarcoma, and high-grade leukemias/lymphomas such as non-Hodgkin lymphoma; these treatment regimens are administered over several days (23-25). Accordingly, we decided to administer a HDMTX treatment to wild-type mice for seven consecutive days in our present investigations as it is in line with the dosage employed in the clinical setting. (pages 15–16, lines 445–453). 

Comment:

7. Some literature citations are not accurate. For example, page 26, lines 561–564, the citation of reference (32) was not associated with the supplementation of Avn-C at a dose of 0.1 g/kg in the diet of rats.

Response:

We are thankful to reviewer for such careful review of our manuscript and for pointing out this error. We have accordingly updated the literature citation in the revised manuscript. 

The concerned paragraph was further changed, and the citation was updated as follows (page 16, lines 474–478): 

In young men and women, oat avenanthramide supplementation has been shown to reduce circulating inflammatory cytokines in addition to inhibiting the production of chemokines and cell adhesion molecules produced by downhill running (35). Furthermore, administration of 10 mg/kg of AVN-C has been reported to prevent hearing loss in normal mice that were exposed to noise and ototoxic drugs (furosemide and kanamycin) (11) (page 16, lines 474–478).

---

## [Decision Letter · Decision Letter 1]

7 Feb 2022

PONE-D-21-29604R1Efficiency of Antioxidant Avenanthramide-C on High-Dose Methotrexate-Induced Ototoxicity in MicePLOS ONE

Dear Dr. Cho

Thank you for submitting your manuscript to PLOS ONE. After careful consideration, we feel that it has merit but does not fully meet PLOS ONE’s publication criteria as it currently stands. Therefore, we invite you to submit a revised version of the manuscript that addresses the points raised during the review process.

The reviewers have noted that your manuscript is much improved but still needs further revision especially in terms of the representation and definition of synapses.

We look forward to receiving your revised manuscript.

Kind regards,

Susan E Shore, Ph.D.

Academic Editor

PLOS ONE

Journal Requirements:

Reviewers' comments:

Reviewer's Responses to Questions

**Comments to the Author**

1. If the authors have adequately addressed your comments raised in a previous round of review and you feel that this manuscript is now acceptable for publication, you may indicate that here to bypass the “Comments to the Author” section, enter your conflict of interest statement in the “Confidential to Editor” section, and submit your "Accept" recommendation.

Reviewer #1: (No Response)

Reviewer #2: (No Response)

2. Is the manuscript technically sound, and do the data support the conclusions?

Reviewer #1: (No Response)

Reviewer #2: Yes

3. Has the statistical analysis been performed appropriately and rigorously? 

Reviewer #1: (No Response)

Reviewer #2: Yes

4. Have the authors made all data underlying the findings in their manuscript fully available?

Reviewer #1: (No Response)

Reviewer #2: Yes

5. Is the manuscript presented in an intelligible fashion and written in standard English?

Reviewer #1: (No Response)

Reviewer #2: Yes

6. Review Comments to the Author

Reviewer #1: The authors have addressed my previous concerns. The quality of the revised manuscript has been improved. However, there are still some issues that need to be addressed.

1. Synaptic ribbons: the authors used the term “synapses” in the manuscript, which is misleading since only presynaptic ribbons (CtBP2) were assessed (as described in the methods section). Synapses indicate both presynaptic ribbons (CtBP2) conjugated with post synaptic terminals (GluA2). The description of the number of presynaptic ribbons in the results section (lines 339–343) is also confusing. In general, the ribbon number should be calculated per inner hair cell, while Fig. 4B, C, and D are labeled as “Synapses/20 µm”. I suggest that the authors review pertinent articles, such as Kohrman et al., 2020, Kujawa and Liberman 2009 and recalculate the presynaptic ribbon number.

2. Fig. 2B, C. y-axes should be labeled as “Threshold (dB SPL)”.

Reviewer #2: The revision has improved the manuscript. There are still some concerns, most around providing methodological details.

Abstract – “remarkable” is not the best descriptor for the threshold shift, it would be expected for an ototoxic effect

Methods – was the total 7 mice in each group or 7 males and 7 females? If the former, what was break-down between males and females. Was this the same in all group? Were different groups of 7 used for each metric (such as detection in body fluids versus hearing tests, scanning EM and hair cell counts, ribbon synapse counting) or was there a total of 7 for all metrics from a condition?

Counting just ribbons and not the combination of ribbon (CTBP2) and receptor (GluR) is not a measure of ribbon synapses. Literature shows there are often ribbons without receptors which would not be a synapse. Ribbon counts are still of value but should be described correctly.

The quantitative analysis of outer hair cell loss needs to be better described. How were the regions counted selected to reduce potential for bias? How were counts done? Similarly how were regions for ribbon counts selected and how were counts done?

7. PLOS authors have the option to publish the peer review history of their article (what does this mean?). If published, this will include your full peer review and any attached files.

Reviewer #1: No

Reviewer #2: No

---

## [Author Response · Author response to Decision Letter 1]

24 Feb 2022

Thank you for giving us the opportunity to submit a revised draft of our manuscript titled “Efficiency of Antioxidant Avenanthramide-C on High-Dose Methotrexate-Induced Ototoxicity in Mice” to PLOS ONE. We truly appreciate the time and effort of the reviewers for providing their valuable feedbacks on our manuscript. We have been able to incorporate relevant changes to reflect most of the suggestions provided by the reviewers. We have highlighted all the changes in the revised manuscript. Response to the reviewers file is attached.

---

## [Decision Letter · Decision Letter 2]

7 Mar 2022

PONE-D-21-29604R2Efficiency of Antioxidant Avenanthramide-C on High-Dose Methotrexate-Induced Ototoxicity in MicePLOS ONE

Dear Dr. Cho,

Thank you for submitting your manuscript to PLOS ONE. After careful consideration, we feel that it has merit but does not fully meet PLOS ONE’s publication criteria as it currently stands. Therefore, we invite you to submit a revised version of the manuscript that addresses the final  points raised during the review process.

We look forward to receiving your revised manuscript.

Kind regards,

Susan E Shore, Ph.D.

Academic Editor

PLOS ONE

Journal Requirements:

Reviewers' comments:

Reviewer's Responses to Questions

**Comments to the Author**

1. If the authors have adequately addressed your comments raised in a previous round of review and you feel that this manuscript is now acceptable for publication, you may indicate that here to bypass the “Comments to the Author” section, enter your conflict of interest statement in the “Confidential to Editor” section, and submit your "Accept" recommendation.

Reviewer #2: (No Response)

2. Is the manuscript technically sound, and do the data support the conclusions?

Reviewer #2: No

3. Has the statistical analysis been performed appropriately and rigorously? 

Reviewer #2: Yes

4. Have the authors made all data underlying the findings in their manuscript fully available?

Reviewer #2: Yes

5. Is the manuscript presented in an intelligible fashion and written in standard English?

Reviewer #2: Yes

6. Review Comments to the Author

Reviewer #2: I have only one question/concern remaining. This is how regions where "squares" for counting ribbons were placed. Was this the same regions in each cochlea assessed. Concern is removing potential for unintentional bias in selection of region to be counted.

7. PLOS authors have the option to publish the peer review history of their article (what does this mean?). If published, this will include your full peer review and any attached files.

Reviewer #2: No

---

## [Author Response · Author response to Decision Letter 2]

13 Mar 2022

Response to the reviewer file is attached

---

## [Editor Report · Decision Letter 3]

15 Mar 2022

Efficiency of Antioxidant Avenanthramide-C on High-Dose Methotrexate-Induced Ototoxicity in Mice

PONE-D-21-29604R3

Dear Dr. Cho,

We’re pleased to inform you that your manuscript has been judged scientifically suitable for publication and will be formally accepted for publication once it meets all outstanding technical requirements.

Kind regards,

Susan E Shore, Ph.D.

Academic Editor

PLOS ONE
---

## [Editor Report · Acceptance letter]

21 Mar 2022

PONE-D-21-29604R3 

*Efficiency of Antioxidant Avenanthramide-C on High-Dose Methotrexate-Induced Ototoxicity in Mice*

Dear Dr. Cho:

I'm pleased to inform you that your manuscript has been deemed suitable for publication in PLOS ONE. Congratulations! Your manuscript is now with our production department. 

Kind regards, 

on behalf of

Dr. Susan E Shore 

Academic Editor

PLOS ONE